# Analysis of Motion Characteristics of Bionic Morphing Wing Based on Sarrus Linkages

**Zhong Yun, Yunhao Feng \*, Xiaoyan Tang**  **and Long Chen**

The College of Mechanical and Electrical Engineering, Central South University, Changsha 410083, China; yunzhong@csu.edu.cn (Z.Y.); txy@csu.edu.cn (X.T.); leon656557@163.com (L.C.)

\* Correspondence: 203711024@csu.edu.cn

**Abstract:** The variant aircraft can flexibly change its shape to achieve the best sports performance under different flight environments and flight missions. Changing the shape of the wings is a major form of variant aircraft. In this paper, the kingfisher was selected as the bionic object to design a morphing wing. A three-stage lateral folding morphing wing based on Sarrus linkages was presented and applied to the construction of an underwater–aerial transmedia aircraft. The multibody dynamics method was used to analyze the kinematic characteristics of the morphing wing and to optimize the design of the folding mechanism for the torque values of the hinges between the bottom links. The results showed that the optimized hinge torque value was reduced to 1.2085 N·m, and the lengths of the bottom two links were calculated to be 140 mm and 142 mm, respectively, based on the optimized results. Finally, a series of wing folding and unfolding motion experiments were conducted to prove that the mechanism enabled the span change smoothly, which verified the rationality of the design of this paper.

**Keywords:** bionic kingfisher; sarrus; morphing wing; motion characteristics; multibody dynamics simulation; design optimization

## 1. Introduction

The bionic morphing wing studied in this paper was suitable for an underwater–aerial transmedia aircraft [1].

The underwater–aerial transmedia aircraft was designed in imitation of a kingfisher's posture, and its corresponding relationship of aerodynamic layout is shown in Figure 1. Figure 1a–d show four typical postures of kingfisher when entering and leaving water. According to the special requirements of large lift force in the air, small resistance in entering water, lowered posture in water, and lift force in coming out of water, our research group designed the stances of aircraft corresponding to kingfisher's flight posture, as shown in Figure 1e–h. The body shape of the aircraft is the streamlined shape of the smooth kingfisher body. The body length was enlarged in proportion to the kingfisher body (the ratio is 1:3.45), and the wingspan was appropriately increased relative to the kingfisher wings to meet the requirements of lift.

The morphing wing is the main form required to realize the variant of the aircraft. A reasonable morphing wing can make the aircraft have better aerodynamic and hydrodynamic performance, improve the adaptability of the aircraft, and increase the endurance of the aircraft [2]. After years of rapid development, there are many types of morphing wings. According to the deformation principle and characteristics, they can be divided into the following categories: telescopic wing, swept-back wing, folding wing, and flexible intelligent variant wing [3,4].

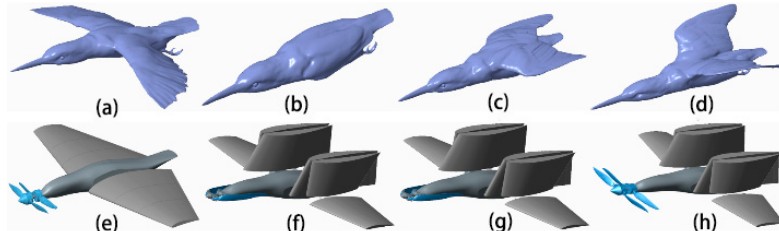

**Figure 1.** Correspondence of aerodynamic layout between aircraft and kingfisher: (**a**) kingfisher flat-flying state; (**b**) kingfisher closing state; (**c**) kingfisher swept back state; (**d**) Kingfisher into the water state; (**e**) aircraft flat-flying state; (**f**) aircraft closing state; (**g**) aircraft swept back state; (**h**) aircraft into the water state.

Different morphing wings have their own advantages, but also have their own difficulties and disadvantages. In the extended state, the retractable wing can play the role of endurance and drag reduction and has good aerodynamic performance. In the retracted state, although the aerodynamic performance is poor, it has better maneuverability. However, the expansion mode will change the position of the aircraft's center of gravity and will increase the size of the aircraft, which is not conducive to underwater navigation [5,6]. The swept wing is a morphing wing that can change its sweep angle. When flying in the air, it can use a larger sweep angle to enhance aerodynamic performance, and when entering water, it can use a smaller sweep angle to make the wingtip close to the fuselage to obtain smaller wingspan and contact area, which can effectively reduce the impact of entering water and the resistance of diving in water. The structure is simple and easy to control. However, larger driving torque is needed at the wing root hinge to achieve the variant, and the wing as a whole is not easy to integrate into the fuselage, so there will still be greater resistance [7–9].

The folding mechanisms used in folding-wing vehicles are mainly: linkage mechanism, gear mechanism, and torsion spring–pulley mechanism. In order to improve the flight performance of collapsible aircraft, Wang, C et al. [10] designed a novel mechanism of bionic foldable wings similar to a beetle's based on the four-plate mechanism theory. Inspired by the beetles, Zhang, Z et al. [11] designed a new type of bionic folding wing for the flapping-wing micro air vehicle (MAV). Ryu, S et al. [12] designed two wing mechanisms based on a four-bar linkage structure: one is only for flapping motion (FM), and the other is for simultaneous flapping and folding motion (FFM) during a wing stroke. Truong et al. [13] develop an artificial foldable wing that mimics the hind wing of a beetle.

This paper proposed a three-stage lateral-folding morphing wing based on Sarrus linkages. The morphing wing is unfolded when flying in the air, with good aerodynamic performance. The wing is folded when entering water and diving, which can effectively reduce the impact force and resistance of water flow. However, large driving torque is needed to drive the folding mechanism, and the impact force sustained at the moment of entering water may directly damage the hinge, which is the most vulnerable part of the whole mechanism [14–16]. Therefore, the force and design optimization of each hinge are also key parts of the morphing wing design in this paper.

The main purpose of the folding bionic morphing wing design in this paper was to realize the water entry and exit process of the underwater–aerial transmedia aircraft. When the aircraft is in the air, the morphing wing is extended; when the aircraft starts to enter the water, the morphing wing begins to fold under the drive of the steering engine; when the aircraft is fully in the water, the morphing wing is fully folded to enable the aircraft to operate underwater. The process of the aircraft coming out of the water is similar to the process of entering the water. The wing is unfolded to achieve the process of exiting the water.

The morphing wing realizes the folding and unfolding of the wing by Sarrus linkages to achieve the water entry and exit processes of the aircraft. The working process of the aircraft is shown in Figure 2.

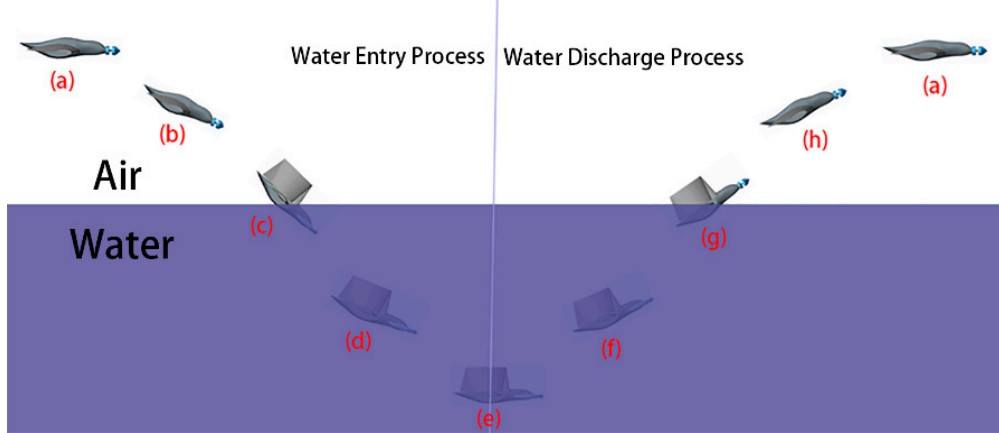

**Figure 2.** Working process of aircraft: (**a**) air navigation process; (**b**) preparation process from air to underwater; (**c**) water entry process; (**d**) underwater attitude adjustment process; (**e**) underwater navigation process; (**f**) preparation process from underwater to air; (**g**) water discharge process; (**h**) aerial attitude adjustment process.

In summary, the main purpose of the folding bionic morphing wing design in this paper was to realize the water entry and exit processes of the underwater–aerial transmedia aircraft in a specified time. Additionally, the impact force sustained at the moment of entering water may directly damage the hinge, so the force and design optimization of each hinge are also key parts of the morphing wing design in this paper.

## 2. Materials and Methods

### 2.1. A Bionic Morphing Wing Based on Sarrus Linkages

The schematic structure of Sarrus linkages is shown in Figure 3, which is a spatial six-bar mechanism with high stiffness, high bearing capacity, simple structure, and low cost [17,18]. It is composed of upper and lower plates (AB) and four side plates (RSTU), which is connected by two groups of six rotating pairs. Without any moving pair or screw pair in the middle, it has relatively high transmission efficiency and is easy to maintain [19,20].

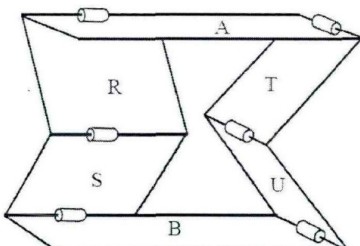

**Figure 3.** Sarrus linkages.

According to the degree of freedom analysis method of spatial closed chain mechanism [21,22], the degree of freedom of the Sarrus mechanism can be obtained, which can be calculated by the following formulas:

$$F = \sum_{i=1}^{P} f_i - \lambda \tag{1}$$

$$\lambda = \lambda_R + \lambda_{PP} + \lambda_{PR} \tag{2}$$

$$F = \sum_{i=1}^{P} f_i - (\lambda_R + \lambda_{PP} + \lambda_{PR}) = 6 - (2 + 0 + 3) = 1 \tag{3}$$

It can be seen that the Sarrus mechanism is a mechanical structure with only one degree of freedom, which can transform the rotation motion of an input terminal into linear motion through the interaction of each member bar in its own mechanism.

Based on the bionics principle and Sarrus mechanism principle, this paper designed the shape of a transmedia aircraft and the structure of a morphing wing. A kingfisher was selected as the bionic object, and the design size of the morphing wing was obtained by equal scale, as shown in Table 1.

**Table 1.** Design dimensions of morphing wing.

| Component | Inner Wing | Middle Wing | Outer Wing | Rod 1 | Rod 2 |
|---|---|---|---|---|---|
| Length/mm | 150 | 150 | 200 | 170 | 180 |

According to the design requirements of this paper, we must choose the wing type with the highest lift to drag ratio; according to the bionic principle, we observe the shape of kingfisher wings and finally choose NACA6412, which possesses a similar shape and a higher lift-to-drag ratio. The shape of NACA6412 wing ribs and their distribution on the Sarrus skeleton are shown in Figure 4.

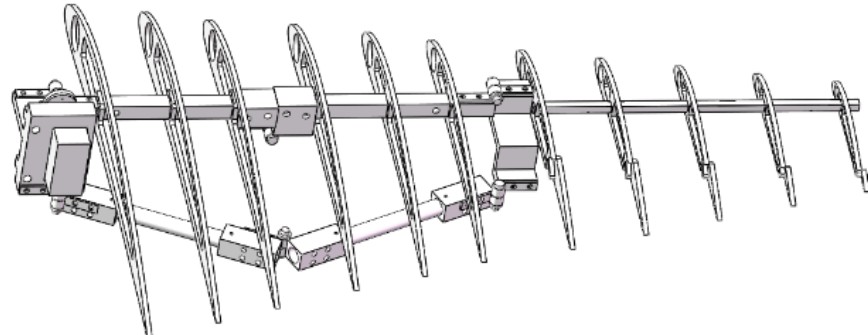

**Figure 4.** The shape of NACA6412 wing ribs and their distribution on the Sarrus skeleton.

SolidWorks software was used for 3D modeling of the morphing wings, as shown in Figure 5. The fuselage is connected to six rods that make up the Sarrus mechanism: the platform, the inner wing, the middle wing, the outer frame, and the bottom rods 1 and 2, of which the inner wing is of equal length to the middle wing, and the rods are connected by hinges.

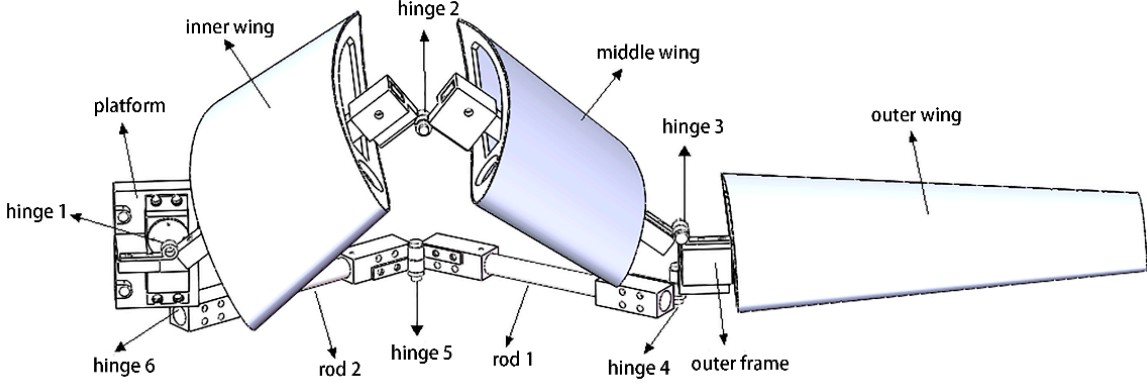

**Figure 5.** Morphing wing based on Sarrus linkages.

According to the principle of the Sarrus mechanism, when the inner wing is driven by the steering engine, the inner wing folds to a certain angle relative to the direction of the wingspan, and the middle wing folds to the same angle along the direction of the wingspan

around hinge 2. Meanwhile, the outer frame is translated in the direction of the wingspan to ensure that the outer wing always keeps level with the body, and the two connecting rods at the bottom move accordingly.

The maximum folding angle of the morphing wings designed in this paper is 90°. Figure 6 shows the expanded and folded states of the morphing wing.

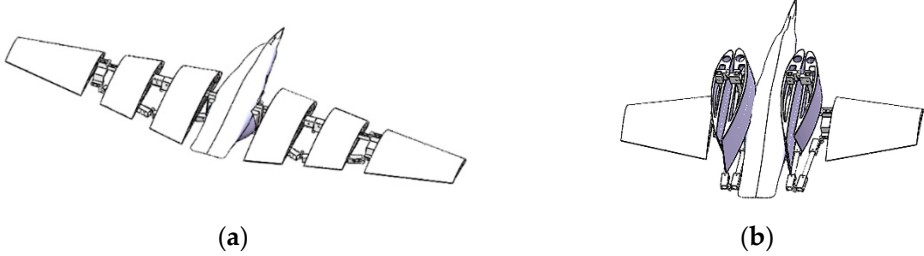

(**a**)　　　　　　　　　　　　　　　　　　　　　　　　(**b**)

**Figure 6.** Diagrammatic sketch of morphing wing: (**a**) morphing wing expanded state; (**b**) morphing wing folded state.

### 2.2. Bionic Morphing Wing Kinematic Model

The morphing wing structure designed in this paper has a degree of freedom of 1, so only one power source is needed to drive it. As can be seen above, the driving object selected in this paper is the inner wing, which means that the rotation angle of the inner wing becomes the input variable of the mechanism.

The center of the left and right wings was taken as the origin of coordinates for analysis. In order to simplify the kinematic analysis of the morphing wing, the motion analysis diagram shown in Figure 7 was made. In the figure, the distance between the fuselage-connecting platform AF and the origin coordinates is $\Delta$. The lengths of the inner wing (AB) and middle wing (BC) are respectively a and b, and the included angles between them and axis X are, respectively, $\alpha_1$ and $\alpha_2$. The lengths of the outer frame and the fuselage connecting platform are both c.

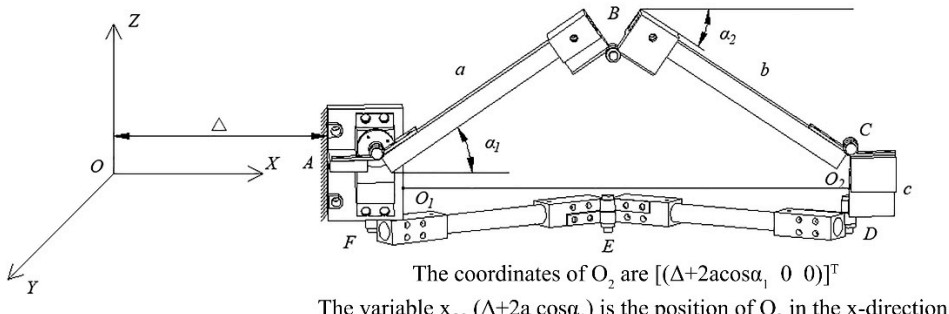

The coordinates of $O_2$ are $[(\Delta + 2a\cos\alpha_1 \quad 0 \quad 0)]^T$
The variable $x_{O_2}$ ($\Delta + 2a\cos\alpha_1$) is the position of $O_2$ in the x-direction

**Figure 7.** Sketch of morphing wing motion analysis.

According to the design requirements, $a = b$, $\alpha_1 = \alpha_2$, the rotational angular velocity, and angular acceleration of the inner wing and the middle wing are the same. The following is the analysis of the motion characteristics of the outer wing. In Figure 5, the distance between the midpoints $O_1$ and $O_2$ of AF and CD is $[\Delta + 2a\cos\alpha_1]$. As can be seen from the figure, their coordinates are, respectively:

$$O_1 = \begin{bmatrix} x_{O_1} & y_{O_1} & z_{O_1} \end{bmatrix}^T = \begin{bmatrix} \Delta & 0 & 0 \end{bmatrix}^T \tag{4}$$

$$O_2 = \begin{bmatrix} x_{O_2} & y_{O_2} & z_{O_2} \end{bmatrix}^T = \begin{bmatrix} \Delta + 2a\cos\alpha_1 & 0 & 0 \end{bmatrix}^T \tag{5}$$

As mentioned above, the morphing wing structure has only one translational degree of freedom along the X-axis, so only one variable is needed to express the position equation of any bar, which can be obtained from the above equation:

$$x_{O_2} = \Delta + 2a\cos\alpha_1 \tag{6}$$

$$\alpha_1 \leq 90°.$$

By taking the first derivative of both sides of the above equation with respect to time $t$, the velocity expression of the outer frame can be obtained:

$$\dot{x}_{O_2} = -2a\dot{\alpha}_1\sin\alpha_1 \tag{7}$$

Continue to take the first derivative of both sides of the above equation with respect to time $t$, and the expression of the acceleration of the outer frame can be obtained

$$\ddot{x}_{O_2} = -2a\ddot{\alpha}_1\sin\alpha_1 - 2a\dot{\alpha}_1{}^2\cos\alpha_1 \tag{8}$$

The above procedure completes the position, velocity, and acceleration analysis of the morphing wing.

### 2.3. Virtual Prototype Folding and Unfolding Motion Simulation

The underwater–aerial transmedia aircraft designed in this paper is unfolded in the air and folded in the water, so the aircraft needs to complete the folding action in the process of diving into the water.

According to the design requirements, the aircraft should fly at least two meters above the water surface. If it falls into the water, according to the free fall formula $\{S = \frac{1}{2}gt^2\}$, the falling time is 0.64 s. Therefore, the folding action of the morphing wing is required to be completed within 0.5 s. According to the movement time requirement of the morphing wing, the steering engine can rotate 90° in 0.5 s time. Therefore, Futaba S9001 was selected as the driving engine model, and its working speed was 0.22 s/60° at 4.8 V.

We imported the 3D model created in SolidWorks into ADAMS, then set the corresponding simulation environment; added gravity, materials, and other basic attributes; created kinematic pairs for the simulation model according to the motion relationship; and made the following assumptions:

(1) The purpose of this simulation and experiment is to verify the feasibility of the folding scheme, so the resistance on the wing during the movement was not considered.
(2) The role of skin is to form the whole wing and provide lift in the process of movement. In order to verify the kinematic characteristics, simulation verification was carried out in static condition in this paper, and the skin weight is light, so the skin structure was removed in the simulation model.
(3) Without considering the elastic deformation of the component, it was regarded as a rigid body.
(4) The frictional resistance between the hinges was ignored.
(5) The virtual prototype model finally established is shown in Figure 8.

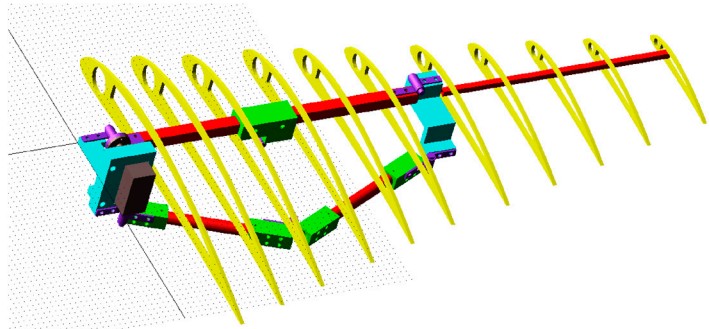

**Figure 8.** Model of unfolded state of morphing wing motion simulation.

The driving part of the virtual model rotated according to the assigned driving function, and the motion function STEP (time, 0, 0 d, 0.33, −90 d) +STEP (time, 2.67, 0 d, 3, 90 d) was added to the driving kinematic pair according to the steering engine data.

The motion simulation of the entire virtual model can be completed by setting the simulation time and step length, and the motion posture of the model at different times can be observed, as shown in Figure 9. The kinematic simulation can obtain the value of each revolute pair force of the morphing wing. At the same time, the key physical parameters such as velocity; displacement; and acceleration of inner, middle, and outer wings can be obtained by motion simulation. Through these data, the design scheme adopted in this paper is fed back to see whether it meets the design requirements.

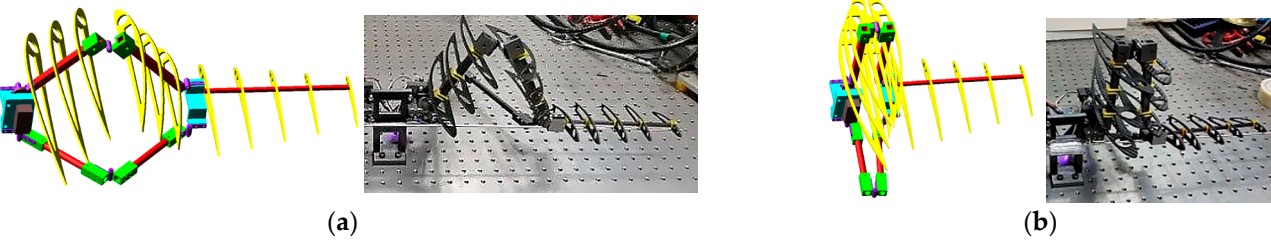

(**a**)                                    (**b**)

**Figure 9.** Comparison of simulation model and prototype folded states: (**a**) folded 45° state; (**b**) folded 90° state.

### 2.4. Prototype Folding and Unfolding Experiment

The morphing wing of the transmedia aircraft is mainly composed of fuselage connecting drive module, wing body, and folding connecting device based on a Sarrus structure. The variation control technology of the morphing wing designed in this paper adopted the electronic control technology, mainly using the control of the electric steering engine, to complete the variation of the wing action. The steering engine is a position servo driver which can continuously change the rotation angle in the process of motion and keep the same motion angle over a certain period of time. It can control the folding angle of the morphing wing.

The fuselage-connecting drive module includes a connecting base, driving–steering engine, and steering engine hinge. Its main function is to connect the wing part with the fuselage and connect the driving–steering engine to this part. The connecting base also bears the weight of the entire wing, so it is strong enough to resist the static bending moment of gravity.

The driving–steering engine is installed on the connecting base. The driving–steering engine is connected with the inner wing connecting rod through the steering actuator hinge, and the output end of the driving–steering engine is connected with the steering engine hinge. The connecting base and steering engine hinge are 3D-printed with nylon material. Futaba S9001 was selected as the driving model of the steering engine. The basic parameters of the steering engine are shown in Table 2:

**Table 2.** Basic parameters of Futaba S9001.

| Size | Weight | Velocity | Torque |
|---|---|---|---|
| 40.4 × 19.8 × 36.0 | 48 g | 0.22 s/60° (4.8 V)<br>0.18 s/60° (6.0 V) | 3.9 kg/cm (4.8 V)<br>5.2 kg/cm (6.0 V) |

The folding connecting device based on the Sarrus structure is the core driving skeleton of the morphing wing designed in this paper and is mainly composed of a wing connecting rod, bottom connecting rod, outer frame, and related connecting pieces (tube clamp, connecting hinge).

The wing carbon tube and the bottom connecting rod are made of carbon tubes according to the design size. The outer frame and the tube clamp are made of nylon 3D printing material. The hinge part connecting the hinge is made of PLA 3D printing material, and the hinge shaft connecting the hinge is replaced by bolts of appropriate size.

In order to verify the feasibility of this morphing wing folding scheme, the author made an experimental prototype according to the size of the simulation model, and the key dimensions and materials of its components are shown in Table 3.

**Table 3.** Production sizes and materials of test prototype.

| Part Name | Inner/Middle Wing | Outer Wing | Rod 1 | Rod 2 | Hinges | Related Connectors |
|---|---|---|---|---|---|---|
| Length/mm | 124 | 200 | 140 | 142 | / | / |
| Material | Square carbon tube | Square carbon tube | Round carbon tube | Round carbon tube | PLA | Nylon |

The attitude sensor was placed at the centroid position of the inner wing connecting rod, and the steering engine was energized to control the angle rotation of the inner wing. The data of time, folding angle, velocity, and acceleration of inner wing movement were recorded in real time by PC software so as to complete the folding–unfolding experiment of the morphing wing. The prototype model and experimental process are shown in Figure 10, and the experimental results were analyzed.

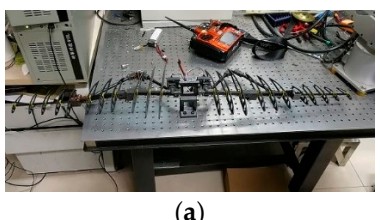 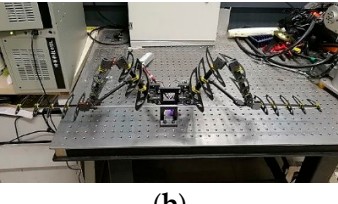 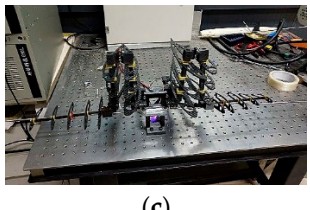 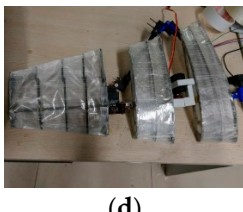

(**a**)  (**b**)  (**c**)  (**d**)

**Figure 10.** Morphing wing experimental prototype: (**a**) prototype unfolded state; (**b**) prototype folded 45° state; (**c**) prototype folded 90° state; (**d**) skinned wings.

## 3. Results

### 3.1. Simulation Motion Analysis and Optimization Design of Morphing Wing

In the process of motion simulation, it can be seen that the morphing wing designed in this paper can smoothly complete folding and unfolding motion. Due to the motion characteristics of the steering engine, the morphing wing can complete folding (unfolding) motion in 0.33 s. The displacement, velocity and acceleration changes of each wing are shown in Figure 11.

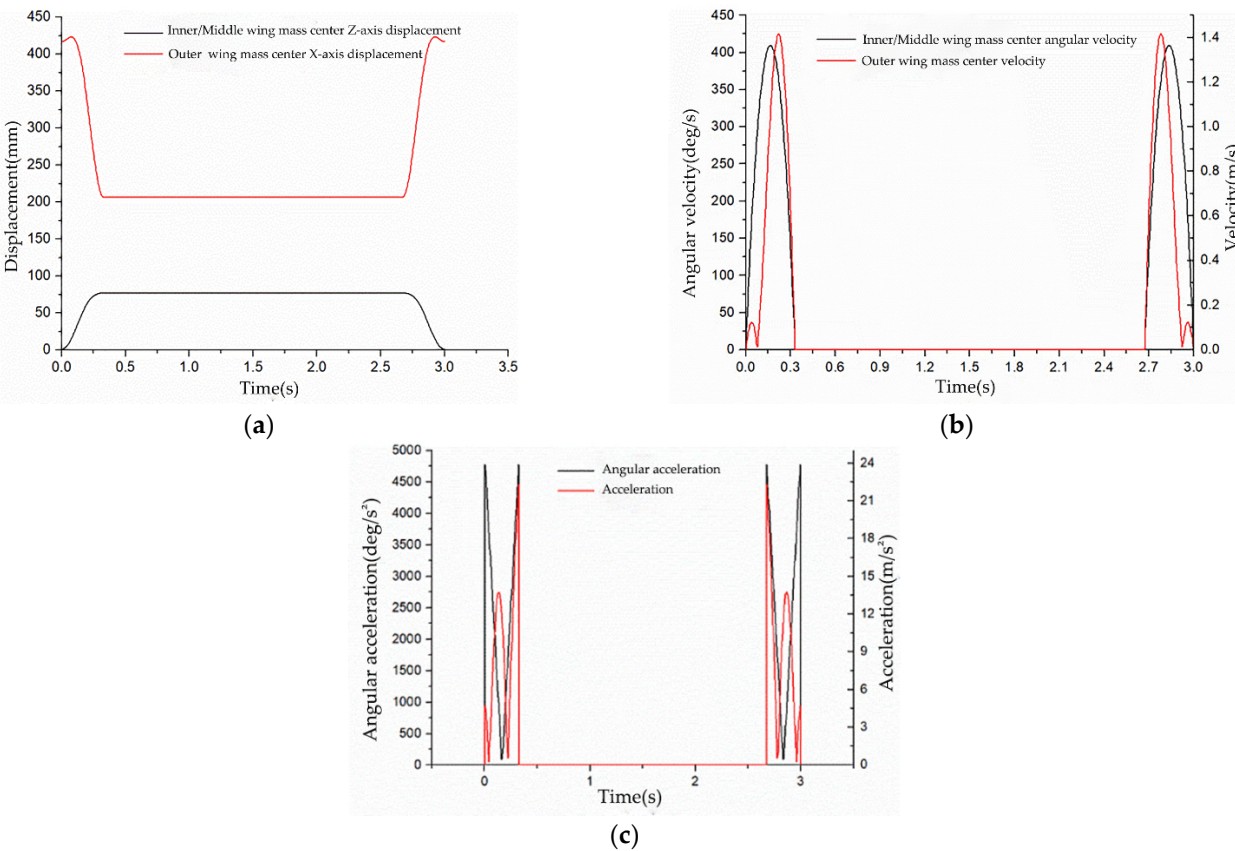

**Figure 11.** Variation of displacement, velocity and acceleration of each wing: (**a**) displacement diagram; (**b**) velocity diagram; (**c**) acceleration diagram.

The skeleton of the morphing wing designed in this paper is a spatial six-bar mechanism based on a Sarrus mechanism. The main bearing point and the most vulnerable part of the structure are the connecting hinges between the links. The measured forces and torques of the six revolute pairs in the model are shown in Figure 12, which can provide help for the strength design of the hinge.

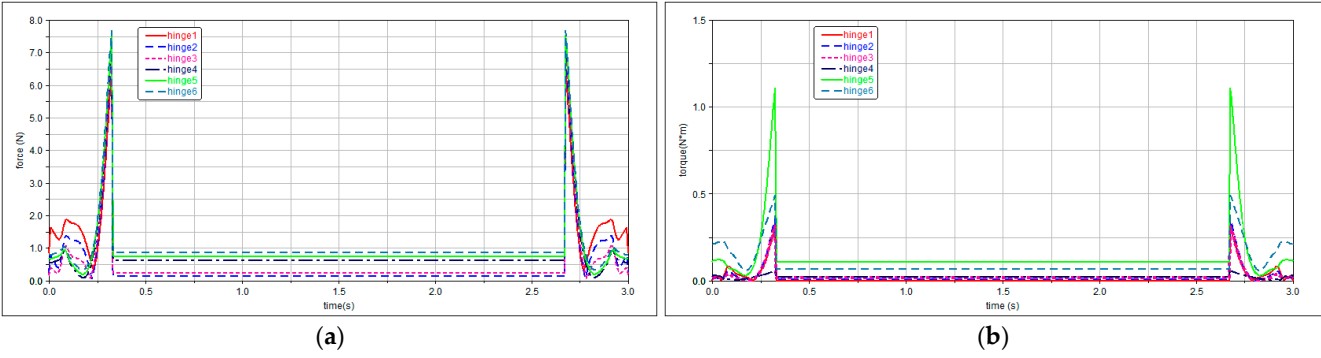

**Figure 12.** Measurement of forces and torques on 6 revolute pairs: (**a**) force on the revolute pairs; (**b**) torque on the revolute pairs.

The hinges are made of PLA 3D printing material, and it can be seen from Figure 12 that all six hinges are subjected to a force of no more than 8 N. In this case, considering the mechanical properties of PLA, all the hinges meet the strength design requirements. As can be seen from Figure 12, the maximum torque value at hinge 1 is less than 0.3 N·m, which translates into a torque of about 3 kg/cm. According to the maximum torque 3.9 kg/cm of

the selected steering engine Futaba S9001, this type of steering engine can provide enough power to make the morphing wing complete its folding action. Additionally, as can be seen from the solid green line in Figure 12, due to the action of gravity, hinge 5—the hinge between the bottom two links—receives the maximum torque value during the movement of all the hinges.

In the design of a transmedia aircraft morphing wing, the strength of hinges is a key aspect that affects the structural strength of the whole wing. On one hand, we can produce high-performance hinges from high-strength materials. On the other hand, we can start from the rod size, because reasonable relative size can make the hinge bear relatively small torque. ADAMS provides powerful parametric modeling and optimization design functions for users to seek the optimal size within the target. Since ADAMS does not support parametric modeling and optimization design of imported models directly, an optimization model must be created in ADAMS first, as shown in Figure 13.

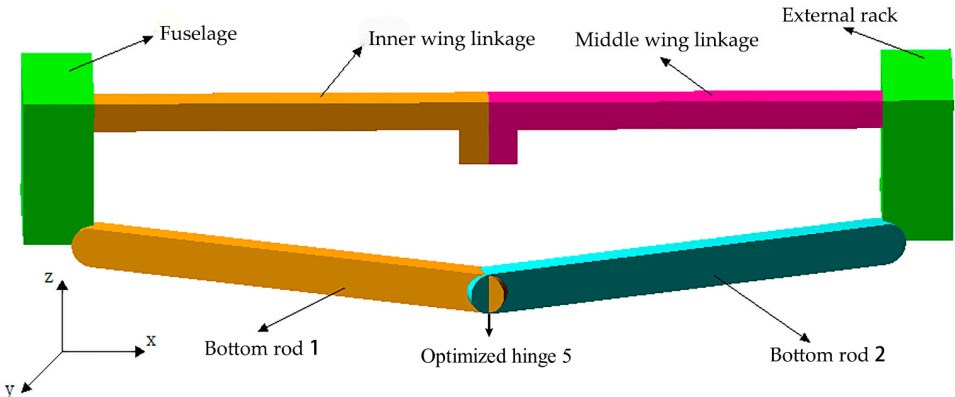

**Figure 13.** Model of ADAMS optimal design.

As shown in the figure above, the parts that do not affect the motion form were removed, and the remaining parts were simplified. Similarly, the same driving function as above was applied to the inner wing connecting rod to make it have the same motion state as the above simulation model. The torque between the bottom two connecting rods (*JOINT_6_MEA_*) was measured, and its value is shown in Figure 14.

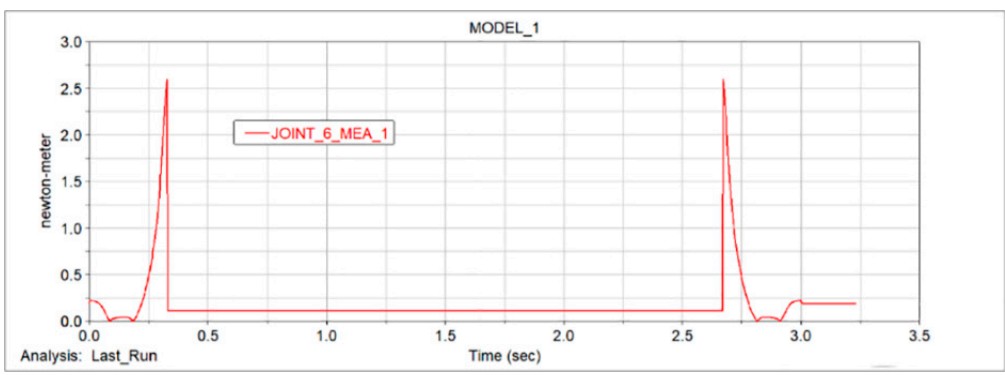

**Figure 14.** Torque of the key hinge of simplified model.

The hinge between the two links at the bottom is the revolute pair subject to the maximum torque, which will then be used as the objective function. The design objective is to minimize the maximum torque value in the process of motion. The body wing designed in this paper was designed in proportion, and the inner wing's connecting rod is equal to the middle wing's connecting rod. Therefore, the variables that can be parameterized in this scheme are the coordinates of the hinge of the bottom two connecting rods. The X-axis coordinates and Y-axis coordinates of this point were parameterized and set as variables *DV_1* and *DV_2*, respectively. The variable *JOINT_6_MEA_1* is the torque value between

the bottom two connecting rods corresponding to the variables *DV_1* and *DV_2*. Since there were only two variables, there was no need to evaluate the design variables, and the optimization design was directly carried out. The optimization analysis results are shown in Table 4.

**Table 4.** Optimization results of design variables.

|  | *JOINT_6_MEA_1* (N·m) | *DV_1* (mm) | *DV_2* (mm) |
| --- | --- | --- | --- |
| Initial Value | 1.5247 | 155 | 45 |
| Optimization value | 1.2085 | 157.21 | 44.97 |

As can be seen from Table 4, the maximum torque suffered by the most critical hinge in the trans-medium aircraft morphing wing based on a Sarrus mechanism is significantly reduced after optimization. Based on the coordinate values 157.21 and 44.97 of the optimized hinge, the lengths of the bottom two links are calculated to be 140 mm and 142 mm, respectively.

### 3.2. Folding and Unfolding Experimental Analysis of Morphing Wing

During the experiment, two groups of folding and unfolding experiments were carried out, and the folding angle of the morphing wing was changed from 0° to 90° successfully. The experimental data in the movement process were exported from the upper computer software. The experimental data of time, velocity, and acceleration of inner wing movement were obtained, and the obtained curves are shown in Figures 15 and 16.

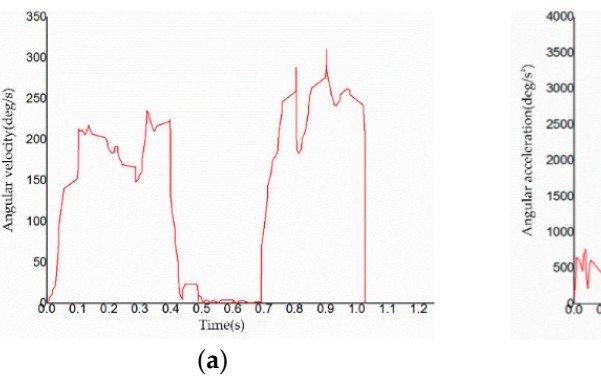
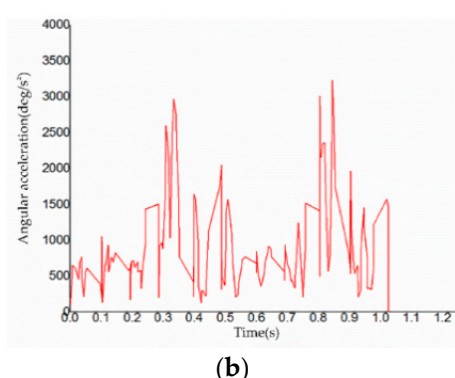

(**a**)  (**b**)

**Figure 15.** First experimental data: (**a**) experimental data graph of inner wing angular velocity; (**b**) experimental data graph of internal wing angle acceleration.

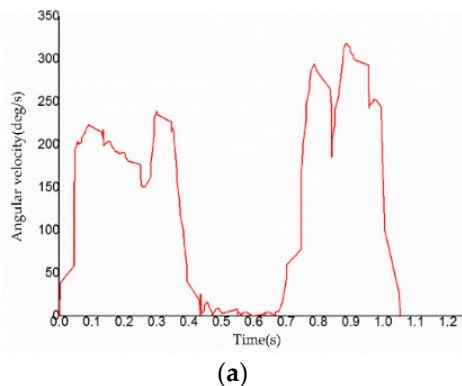
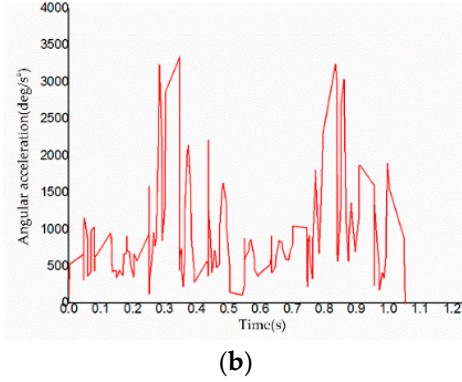

(**a**)  (**b**)

**Figure 16.** Second experimental data: (**a**) experimental data graph of inner wing angular velocity; (**b**) experimental data graph of internal wing angle acceleration.

According to the curves, it is found that the variation trend of the two experimental data is similar, and the movement velocity peak of the inner wing is close to the acceleration

peak, but the curve fluctuates violently. This is due to the inevitable shaking and its own vibration phenomenon during the movement of the rod.

In this paper, the folding and unfolding motion simulation of the morphing wing's folding mechanism was firstly carried out, and the inner wing movement velocity and acceleration diagram as shown in Figure 11 was obtained. By comparing the curves of experimental data with those of simulation data, it was found that the variation trend of the angular velocity of the inner wing during the experiment was consistent with the simulation results, which firstly increased to the peak value, immediately accelerates to 0, and then remained unchanged. The acceleration of inner wing movement also had such a movement trend that it increased to the peak value quickly, decreased rapidly, increased to the peak value rapidly, and then decreased to zero and remained unchanged.

During the experiment, the upper computer software found that in the initial state, due to gravity, the wing part sagged relative to the fixed frame, so the folding angle was not zero relative to the horizontal plane, but about one or two degrees negative.

The rocker of the remote control was moved to make the morphing wing fold and unfold reciprocally. Meanwhile, the relative angle of the folded wings to the maximum position was observed in the upper computer software. The folding angles of the morphing wing in the two experiments are shown in Tables 5 and 6.

**Table 5.** Folding angle of the morphing wing in the first experiment.

| Time/s | 0 | 0.4378 | 0.6915 | 1.027 |
|---|---|---|---|---|
| Folding angle/° | −1.1755 | 86.1909 | 86.2036 | −1.1646 |

**Table 6.** Folding angle of the morphing wing in the second experiment.

| Time/s | 0 | 0.431 | 0.7005 | 1.054 |
|---|---|---|---|---|
| Folding angle/° | −1.593 | 86.3228 | 86.1759 | −1.3458 |

For the first experiment, the folding angle of the morphing wing was $-1.1755°$ at the relative time of 0 s, and the folding angle gradually increased with time and finally reached a peak at the relative time of 0.437 s—when the folding angle was $86.1909°$—and the folding angle remained at $86.1909 \pm 0.2°$ while the controller rocker remained motionless. Then, after 0.6915 s relative time, the folding angle gradually decreased, i.e., the morphing wing was performing the unfolding motion; finally, at 1.027 s relative time, the folding angle became $-1.1646°$, and remained at this folding angle with an error of $0.2°$ for the rest of the time. The situation of the second experiment is similar to the first experiment, and the characteristic time and corresponding angle are shown in Table 6.

It was found that the maximum folding angle of the morphing wing is between $85°$ and $87°$ due to gravity and the errors in the prototype-manufacturing process. Therefore, it is concluded that the morphing wing prototype can complete the transition from its unfolded state to its folded state.

The experimental data were then analyzed to determine how long it took the morphing wing to complete a single action, that is, the time it took for a steering gear to turn through its entire range. Recording the first experiment, it can be seen from several key relative time points in the experimental data that the morphing wing took 0.437 s to complete the folding motion and 0.3355 s to complete the unfolding motion. Recording the second experiment, it can be seen that the morphing wing took 0.431 s to complete the folding motion and 0.3535 s to complete the unfolding motion.

Through the two groups of experiments, it was found that the folding time of the wings is about 0.434 s, the unfolding time is about 0.3445 s, and the theoretical time of the folding or unfolding action is 0.33 s. The relative errors between the theoretical time (0.33 s) and experimental data of the folding and unfolding actions are 31.5% and 4.39%. This is because the morphing wing is affected by gravity, and the outer wing part sags slightly

compared with the inner wing part during the experiment. This is also why the time of the folding action is about 0.09 s more than that of unfolding. In addition, friction and other factors are ignored in the simulation process, so the experimental time will be more than the theoretical time, and the time error is within the allowable range of the actual situation. Therefore, the validity of the simulation calculation and the rationality of the scheme design are verified to some extent by the unfolding and folding experiment of the morphing wing.

## 4. Conclusions

The underwater–aerial transmedia aircraft morphing wing designed in this paper could precisely complete the folding action and compress the space of the overall structure as much as possible. The Sarrus linkage acts as a spatial parallel linkage mechanism; its high strength ensured the stability of the morphing wing during the process of movement. Through the two groups of experiments, it was found that the folding time of the wings is about 0.434 s, the unfolding time is about 0.3445 s, and the time of the design requirements is 0.5 s. The folding and unfolding actions in the experiments have effectively taken less time than the design requirements to complete them. In summary, the morphing wing designed by selecting the kingfisher as the bionic object can be better applied to a specific type of underwater–aerial transmedia aircraft.

When the Sarrus mechanism has dual stability, it has two equilibrium positions, which are precisely adapted to the two states of the underwater–aerial transmedia aircraft morphing wing. Although a rigid hinge was used in this paper, the steady-state characteristics of the flexible Sarrus mechanism can provide ideas for the design of the morphing wing later. Additionally, the operating range of the underwater–aerial transmedia aircraft designed in this paper was limited to low-altitude subsonic speed. We have not analyzed the wing motion characteristics under other operating conditions in depth, but this is the direction of our subsequent work. For the question about the heavy crosswind load, we consider the underwater–aerial integrated propulsion system to control the tilt of the fuselage to adapt to the wind direction to solve this problem. As for the pitch, roll, and yaw of the aircraft, there is no relevant study in this paper, but it is an important part of our future research.

**Author Contributions:** Conceptualization, Z.Y. and Y.F.; methodology, X.T. and L.C.; software, Z.Y. and Y.F.; validation, Z.Y., X.T., Y.F. and L.C.; formal analysis, Z.Y. and Y.F.; investigation, Z.Y. and Y.F.; resources, Z.Y.; data curation, Y.F. and L.C.; writing—original draft preparation, Y.F. and L.C.; writing—review and editing, Z.Y. and X.T.; visualization, Y.F.; project administration, Z.Y. All authors have read and agreed to the published version of the manuscript.

**Funding:** This research was supported by the Natural Science Foundation of Changsha City, China (Grant No. 2022kq2202074) and the Natural Science Foundation of Hunan Province, China (Grant No. 2022JJ30696).

**Institutional Review Board Statement:** Not applicable.

**Informed Consent Statement:** Informed consent was obtained from all subjects involved in the study.

**Data Availability Statement:** Not applicable.

**Conflicts of Interest:** The authors declare no conflict of interest.

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
