# Peer review of "Analysis of Motion Characteristics of Bionic Morphing Wing Based on Sarrus Linkages"

_applsci, doi:10.3390/app12126023_

Round 1

Reviewer 1 Report

Please find attached comments for the review of the following paper ‘’Analysis of Motion Characteristics of Bionic Morphing Wing Based on Sarrus Linkages’’.

We will recommend this paper for publication with the minor revisions.

Reviewer 2 Report

The paper on bionic morphing wing based on Sarrus Linkages is written well. Concepts and results are clearly presented.

Author Response

Thank you very much for your recognition of our work.

Reviewer 3 Report

Dear Authors,

The manuscript is a very good effort on reducing the high torque for morphing wing applications and smooth span change mechanism. It is an interesting read for the Journal readers. However, please answer these questions to have a better understanding of the study for the readers.

1) What is the mission of this design? As in, when it goes underwater is it decreasing the payload or keeping the same? What is the effect of different payload on this mechanism?

2) How did you decide the wing shape? Was there any basic lift to drag ratio optimization performed? 

3) How does this scale?

4) In a scenario of heavy cross wind-load how does this mechanism behave?

5) Where would be the battery in the system and how does pitch, roll, yaw come in to play here?

Please answer these questions in the manuscript to be considered for publishing with this interesting topic.
